# Environmental Correlates to Genetic Diversity and Structure in Invasive Apple Snail (*Pomacea canaliculata*) Populations in China

Xiongjun Liu [1,2,†], Yu Zhou [3,†], Shan Ouyang [2] and Xiaoping Wu [2,*]

1   Guangdong Provincial Key Laboratory of Conservation and Precision Utilization of Characteristic Agricultural Resources in Mountainous Areas, School of Life Sciences, Jiaying University, Meizhou 514015, China; 202001173@jyu.edu.cn
2   School of Life Sciences, Nanchang University, Nanchang 330031, China; ouys1963@ncu.edu.cn
3   Zunyi Academy of Forestry, Zunyi 563000, China; 15585230087@163.com
*   Correspondence: xpwu@ncu.edu.cn
†   These authors contributed equally to this work.

**Abstract:** Invasive species are one of the most serious threats to biodiversity. *Pomacea canaliculata* is considered one of the world's 100 worst invasive species. Major determinants of invasive species distribution are their environmental tolerances, and an understanding of correlations between local environmental variables (e.g., pH, concentration of dissolved oxygen) and genetic diversity is necessary to better prevent and manage the spread of invasive species. However, while such studies have demonstrated associations between the distribution and density of *P. canaliculata* and water quality correlates, the principal mechanisms relating genetic and these environmental correlates have not been fully articulated. Here, the correlation between physicochemical parameters and genetics of *P. canaliculata* were analyzed. The results showed that *P. canaliculata* among the six collection locations had robust genetic diversity, significant genetic differentiation, limited gene flow, and stable population dynamics. RDA analysis showed that genetic variation in *P. canaliculata* was significantly correlated with concentration of dissolved oxygen and pH. These results will provide a basis for effectively preventing and managing the spread of invasive species and identifying which habitats may be more at risk of invasion.

**Keywords:** genetic diversity; invasive species; water quality; biodiversity loss





## 1. Introduction

Freshwater ecosystems are under multiple severe anthropogenic threats, such as water pollution and biological invasions [1,2]. Biological invasion is one of the main factors that contribute to the loss of biodiversity and function of ecosystems in freshwater habitats [3,4]. Invasive species also seriously affect social and economic development and human health [5]. Invasive freshwater gastropods are among the most successful groups of invasive species in aquatic ecosystems, with the capacity to colonize a wide range of aquatic environments as a consequence of broad physiological tolerances to multiple environmental conditions [6–8]. Many invasive gastropods are still expanding their range, and early detection and prediction of distribution are important for the effective management of such invasive species to minimize negative impacts [9–11].

The golden apple snail, *Pomacea canaliculata* (Lamarck, 1819), is a large freshwater gastropod (maximum shell length of 20 cm; maximum weight of 600 g) native to South America that has spread rapidly outside of its native range in recent decades [7,12]. *P. canaliculata* poses a threat to agricultural crops by gnawing seedlings in their growth period, causing direct economic losses [7,12,13]. *P. canaliculata* may also drive declines of native snails [10,11] and major shifts in ecosystem state and function [14]. Because of rapid social and economic

development, there has been a dramatic increase in the protein content in human diets [15]. As a result of this need for higher protein diets, *P. canaliculata* were initially widely promoted as human food and as a protein supplement in animal feed [12]. However, as apple snails are a regular host of the nematode (*Angiostrongylus cantonensis* Chen, 2935) responsible for eosinophilic meningitis [16,17], aquaculture of *P. canaliculata* for human consumption was unsustainable. Abandoned snails from closed facilities, however, soon established large populations in many freshwater habitats, dispersing through passive transportation via water flow in drainage systems and active dispersal via crawling [12,16,18]. *P. canaliculata* have been introduced to North America [19,20], Europe [21], and Asia [15,22]. *P. canaliculata* were initially introduced to Zhongshan, Guangdong province, in the mainland of China in 1981 through aquaculture aimed at producing snails for human consumption, and the distribution of this species has now expanded northward as a result of rapid environmental change [15,23]. *P. canaliculata* has therefore been classified as one of the world's 100 worst invasive species [24], and the species was also listed as the first of 16 most impactful invasive alien species by the Chinese Environmental Protection Bureau in 2003. However, their ecological impacts are more difficult to estimate, as they also continue to spread into nonagricultural wetlands of many countries [15,19].

The habitat characteristics of invasive species play key roles in determining colonization success [20]. Abiotic conditions of a system may prevent establishment or allow it to thrive, reproduce, and become invasive. For freshwater snails, key abiotic water quality variables are likely to be the concentration of calcium carbonate [25,26], pH [9,20], concentration of dissolved oxygen [27], salinity [28,29], and temperature [30,31]. Previous studies on the invasion of *P. canaliculata* mainly focused on distribution [19], associated diseases [32,33], phylogeny [15,19,22], and taxonomy [34]. However, relating water quality variables to metrics of genetic diversity of *P. canaliculata* in China has not been studied to date. Introduced populations that are established by a few individuals usually experience genetic bottlenecks, and genetic variability is expected to decrease in the new range of colonization [35]. Previous work has suggested that the apple snail has been introduced in China multiple times either through anthropogenic means or secondary spread [15]. Evidence for this is seen with different haplotypes within the same site as well as different combinations of haplotypes among sites within the system [15]. Therefore, knowledge of the genetic diversity and structure of invasive species in non-native areas provides insights into possible control and management of the process of biological invasion [36]. Here, we analyzed the correlation between a suite of physicochemical parameters and the genetic diversity of *P. canaliculata*. This study will provide valuable information for guiding effective prevention, management, and spread of invasive *P. canaliculata* in Chinese freshwater ecosystems.

## 2. Material and Methods

### 2.1. Sample Collection and DNA Extraction

A total of 100 specimens of *P. canaliculata* were collected from six tributary rivers of Poyang Lake, China, in 2016 and 2017 (Table 1; Figure 1). The six collection locations were the Gongshui River (23 individuals), the Tao River (10 individuals), the Shangyou River (6 individuals), the Shushui River (25 individuals), the Gan River (12 individuals), and the Xinjiang River (23 individuals; Table 1; Figure 1). Tissues of individual specimens were preserved in 95% ethanol and stored at −20 °C until DNA extraction. Whole specimens were deposited in the Museum of Biology, Nanchang University (NCUMB). The TINAamp Marine Animals DNA Kit (Tiangen, Beijing, China) was used to extract genomic DNA from mantle tissue, and a Nanodrop 2000 (Thermo Scientific, Beijing, China) and agarose gel electrophoresis were used to estimate the concentration and quality of the extracted DNA.

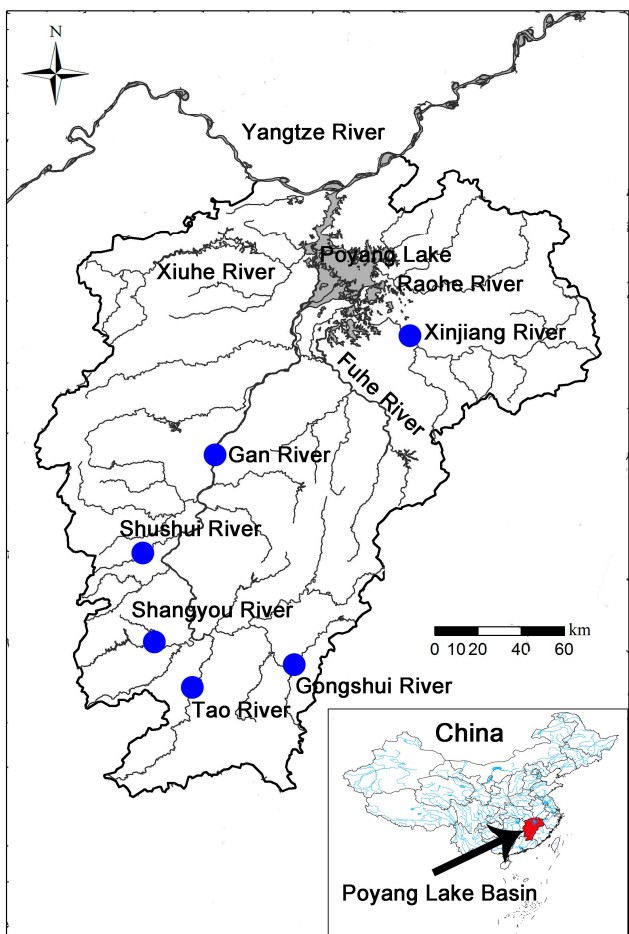

**Figure 1.** Collection locations of *P. canaliculata*.

## 2.2. PCR Amplification

The cytochrome c oxidase subunit-I (COI) primers [the forward primer sequences were LCO1490 (5′-AGTTTACTTATTCGTGCTG-3′), and the reverse primer sequences were HCO2198 (5′-GTATTAAAATTTCGATCAGT-3′)] [37] and 16S rRNA primers [the forward primer sequences were 16SF (5′-CGCCTGTTTAAC AAA AAC AT-3′), and the reverse primer sequences were 16SF (5′-CGCCTGTTTAAC AAA AAC AT-3′)] [38] were used in PCR amplification. The PCR reaction for COI and 16S rRNA primers was performed in 25 μL and 15 μL volumes, including 12.5 μL and 7.5 μL 2× Taq PCR Master Mix (TianGen), 9.5 μL and 6.2 μL ddH$_2$O, 1.0 μL and 0.3 μL 10 μM forward primer, 1.0 μL and 0.3 μL 10 μM reverse primer, and 1 μL and 0.7 μL genomic DNA (about 100 ng/μL), respectively. PCR amplifications for COI primer and 16S rRNA primers were conducted with the following touchdown thermal cycling program: an initial denaturation at 94 °C for 3 min, followed by 34 cycles of 94 °C for 1 min and 35 cycles of 94 °C for 1 min, annealing temperature of 50 °C for 30 s and 53 °C for 1 min, 72 °C for 50 s and 72 °C for 1 min, and a final extension at 72 °C for 8 min and 72 °C for 7 min, respectively. To confirm successful amplification, a 1% agarose gel was used to electrophorese the PCR products, and they were purified with an EZ-10 Spin Column PCR Product Purification Kit (Promega, Madison, WI, USA). In addition, an ABI 3730XL DNA Analyser (Applied Biosystems, Carlsbad, CA, USA) was used to sequence the purified DNA.

## 2.3. Data Analysis

COI sequences were aligned by MUSCLE [39] in MEGA7 according to the translated amino acid sequences, and 16S rRNA gene sequences were aligned in MAFFT v7 [40] with the Q-INS-i algorithm. The two (COI and 16S) datasets were concatenated for phy-

logenetic analyses using SequenceMatrix [41]. The phylogeny of the *P. canaliculata* was performed using Bayesian inference in MRBAYES v.3.2.2 [42]. The initial models of evolution of the COI dataset (HKY+G), 16S dataset (GTR+I+G) and COI+16S dataset (GTR+I+G) were determined using MRMODELTEST v.2.2. A total of 3 million generations, 6 concurrent Markov chains, and 2 hot chain sampling intervals of 100 generations for a total of 30,000 trees were used to run MRBAYES. The stability of log-likelihood is guaranteed by 25% (7500 trees) [43]. The COI and 16S sequences available in GenBank from *P. maculata*, *Pomacea paludosa*, *Pomacea diffusa*, and *Pila conica* were used as outgroups.

DNASP 5.0 [44] was used to analyze the number of haplotypes (*H*), haplotype diversity ($H_d$), and nucleotide diversity ($\pi$) for each *P. canaliculata* population from COI, 16S, and COI+16S datasets. TCS 1.21 [45] was used to construct a haplotype network of COI, 16S, and COI+16S datasets for freshwater mussels with a threshold of 95% and plotted using tcsBU [46].

A hierarchical analysis of molecular variance (AMOVA) was used to evaluate the genetic structure in the COI, 16S, and COI+16S datasets based on ARLEQUIN 3.5 [47]. AMOVA was used to analyze to test the significance of each pairwise population comparison based on 1000 permutations. Tajima's D and Fu's Fs tests were performed by ARLEQUIN 3.5 [47]. The mismatch distribution analysis (MDA) was analyzed using DNASP 5.0 [44]. The Bayesian Skyline Plot (BSP) was analyzed using BEAST 1.4.7 and TRACER 1.5 [48,49]. We used the (GTR+I+G) substitution model from JMODELTEST 2 [50] analysis according to the Akaike information criterion (AIC). The MDA-derived expansion parameter (*Tau*) was converted to absolute estimate of time by *Tau* = 2 *μkt* (each generation time of 2 years, the length of the sequence (*k*), and the mutation rate $\mu = 2.0 \times 10^{-8}$) [51–53], with the assumption of a conventional mitochondrial molecular clock.

*2.4. Correlation between Genetic Diversity and Physicochemical Measurements*

Eleven physicochemical parameters at six rivers during October 2016 and January, May, and August 2017 were measured four times in the study area (Table S1). The water quality data come from Xing et al. (2019) and Li et al. (2019) [54,55]. A YSI 650MDS (YSI) multiparameter meter was used to measure water temperature (°C), dissolved oxygen concentration (mg/L), pH, turbidity (NTU), total dissolved solids (mg/L), nitrate nitrogen (mg/L), and ammonium ($NH_4^+$) concentration (mg/L). A chlorophyll meter (PCH-800, made by the Probest company of Fuzhou City, China) was used to measure chlorophyll-a concentration (mg/L). In addition, we used concentrated $H_2SO_4$ to preserve 600 mL water samples from four sampling periods. These samples were then refrigerated and transported to the Nanchang University laboratory. Total nitrogen (TN) and total phosphorus (TP) were analyzed by ultraviolet spectrophotometry according to Wei et al. (1989) and Huang et al. (1999) [56,57].

The river habitat was analyzed using principal coordinate analysis (PCoA), based on physicochemical parameter data, via the "VEGAN" and "CMDSCALE" packages implemented in R 3.2.0 [58]. To evaluate correlations between water quality and genetic variation of *P. canaliculata*, a redundancy analysis (RDA) was performed by CANOCO 4.5 [59,60]. The 999 Monte Carlo permutation tests were used to analyze the variance (*p* < 0.05) of the RDA gradient. All water quality variables and genetic diversity metrics were log10(X + 1) transformed to meet the assumptions of multivariate normality and to moderate the influence of extreme data [61]. Mantel tests [62] performed in R [58] using the VEGAN package were used to assess the correlations between the matrices of genetic diversity and the matrices of water quality.

## 3. Results

*3.1. Genetic Diversity of P. canaliculata*

Six COI haplotypes were identified from the 100 sequenced *P. canaliculate* individuals from six collection populations (Table 1). The Xinjiang River had the greatest variation with six haplotypes, and the Tao, Shangyou, and Shushui rivers had the lowest variation

with two haplotypes (Table 1). Haplotype diversity values of *P. canaliculata* varied between 0.333 and 0.787 (Table 1). The greatest haplotype diversity was at the Xinjiang River (0.787). Nucleotide diversity values of *P. canaliculata* ranged from 0.007 to 0.031 (Table 1). The Xinjiang River (0.031) had the greatest nucleotide diversity (Table 1). The Shangyou River had the lowest haplotype diversity and nucleotide diversity (Table 1).

Seven 16S haplotypes were identified from the 84 sequenced *P. canaliculata* individuals from six collection populations (Table 1). The Gongshui, Shushui, and Xinjiang rivers had the greatest variation with three haplotypes (Table 1). Haplotype diversity values of *P. canaliculata* varied between 0 and 0.522 (Table 1). The greatest haplotype diversity was at the Xinjiang River (0.522). Nucleotide diversity values of *P. canaliculata* ranged from 0 to 0.004 (Table 1). The Xinjiang River (0.004) had the greatest nucleotide diversity (Table 1). The Tao, Shangyou, and Gan rivers had the lowest haplotype diversity and nucleotide diversity (Table 1).

Fifteen COI+16S haplotypes were identified from the 84 sequenced *P. canaliculata* individuals from six collection populations. The Xinjiang River had the greatest variation with ten haplotypes (Table 1). Haplotype diversity values of *P. canaliculata* varied between 0 and 0.904 (Table 1). The greatest haplotype diversity was at the Xinjiang River (0.904). Nucleotide diversity values of *P. canaliculata* ranged from 0 to 0.019 (Table 1). The Xinjiang River (0.019) had the greatest nucleotide diversity (Table 1). The Shangyou River had the lowest haplotype diversity and nucleotide diversity (Table 1).

**Table 1.** Collection locations, genetic diversity, and neutrality tests of *P. canaliculata* based on COI, 16S rRNA, and COI+16S rRNA datasets. $N$: number of individuals, $H$: number of haplotypes, $H_d$: haplotype diversity, $\pi$: nucleotide diversity.

| Collection Locations | COI Datasets | | | | | | 16S Datasets | | | | | | COI+16S Datasets | | | | | |
|---|---|---|---|---|---|---|---|---|---|---|---|---|---|---|---|---|---|---|
| | Genetic Diversity | | | | Neutrality Tests | | Genetic Diversity | | | | Neutrality Tests | | Genetic Diversity | | | | Neutrality Tests | |
| | $N$ | $H$ | $H_d$ | $\pi$ | Tajima's D | Fu's Fs | $N$ | $H$ | $H_d$ | $\pi$ | Tajima's D | Fu's Fs | $N$ | $H$ | $H_d$ | $\pi$ | Tajima's D | Fu's Fs |
| Gongshui River | 23 | 3 | 0.530 | 0.023 | 2.11 | 17.99 | 21 | 3 | 0.186 | 0.001 | −1.51 | −1.91 | 21 | 4 | 0.605 | 0.013 | 2.64 | 13.67 |
| Tao River | 10 | 2 | 0.556 | 0.025 | 2.73 | 14.08 | 6 | 1 | 0 | 0 | 0 | 0 | 6 | 2 | 0.533 | 0.014 | 1.37 | 9.47 |
| Shangyou River | 6 | 2 | 0.333 | 0.007 | −1.45 | 4.82 | 5 | 1 | 0 | 0 | 0 | 0 | 5 | 1 | 0 | 0 | 0 | 0 |
| Shushui River | 25 | 2 | 0.520 | 0.024 | 3.58 | 23.39 | 25 | 3 | 0.157 | 0.001 | −1.51 | −2.12 | 25 | 4 | 0.597 | 0.014 | 3.16 | 15.99 |
| Gan River | 12 | 3 | 0.667 | 0.026 | 1.85 | 12.21 | 10 | 1 | 0 | 0 | 0 | 0 | 10 | 2 | 0.533 | 0.014 | 2.43 | 13.80 |
| Xinjiang River | 23 | 6 | 0.787 | 0.031 | 2.02 | 12.53 | 17 | 3 | 0.522 | 0.004 | 1.01 | 2.20 | 17 | 10 | 0.904 | 0.019 | 1.65 | 3.06 |
| Total | 100 | 6 | 0.664 | 0.028 | | | 84 | 7 | 0.331 | 0.002 | | | 84 | 15 | 0.697 | 0.016 | | |

### 3.2. Genetic Structure of P. canaliculata

The phylogenetic analysis of COI, 16S, and COI+16S datasets showed that the differences among populations were shallow and that the genetic structure among *P. canaliculata* haplotypes was not well resolved (Figure 2). The phylogenetic analysis of COI, 16S, and COI+16S datasets showed that the clade of *P. canaliculata* had strong posterior probability support (1.00, 0.85, 1.00; Figure 2A,C,E). Six COI haplotypes formed three clades: three haplotypes from the Gongshui River, Tao River, Shangyou River, Shushui River, Gan River, and Xinjiang River formed the first clade; two haplotypes from the Gongshui River, Tao River, Shushui River, Gan River, and Xinjiang River formed the second clade; and one haplotype from the Xinjiang River formed the third clade (Figure 2A). Seven 16S haplotypes formed one clade (Figure 2C). Fifteen COI+16S haplotypes formed three clades: eight haplotypes from the Gongshui River, Tao River, Shushui River, Gan River, and Xinjiang River formed first clade; five haplotypes from the Gongshui River, Tao River, Shangyou River, Shushui River, Gan River, and Xinjiang River formed the second clade; and two haplotypes from Xinjiang River formed the third clade (Figure 2E).

The haplotype network of COI, 16S, and COI+16S datasets for *P. canaliculata* showed similar geographic structure to the phylogenetic analysis and similarly a general lack of geographic resolution (Figure 2). The most frequent COI haplotype of *P. canaliculata* (Hap2) had 41 individuals shared from five collection populations, and two COI haplotypes (Hap5; Hap6) were rare haplotypes (Figure 2B). The most frequent 16S haplotype of *P. canaliculata* (Hap1) had 68 individuals shared from six collection populations, and five 16S haplotypes

(Hap2; Hap3; Hap4; Hap5; Hap7) were rare haplotypes (Figure 2D). The most frequent COI+16S haplotype of *P. canaliculata* (Hap1) had 37 individuals shared from six collection populations, and ten COI+16S haplotypes (Hap3; Hap4; Hap5; Hap6; Hap9; Hap11; Hap12; Hap13; Hap14; Hap15) were rare haplotypes (Figure 2F).

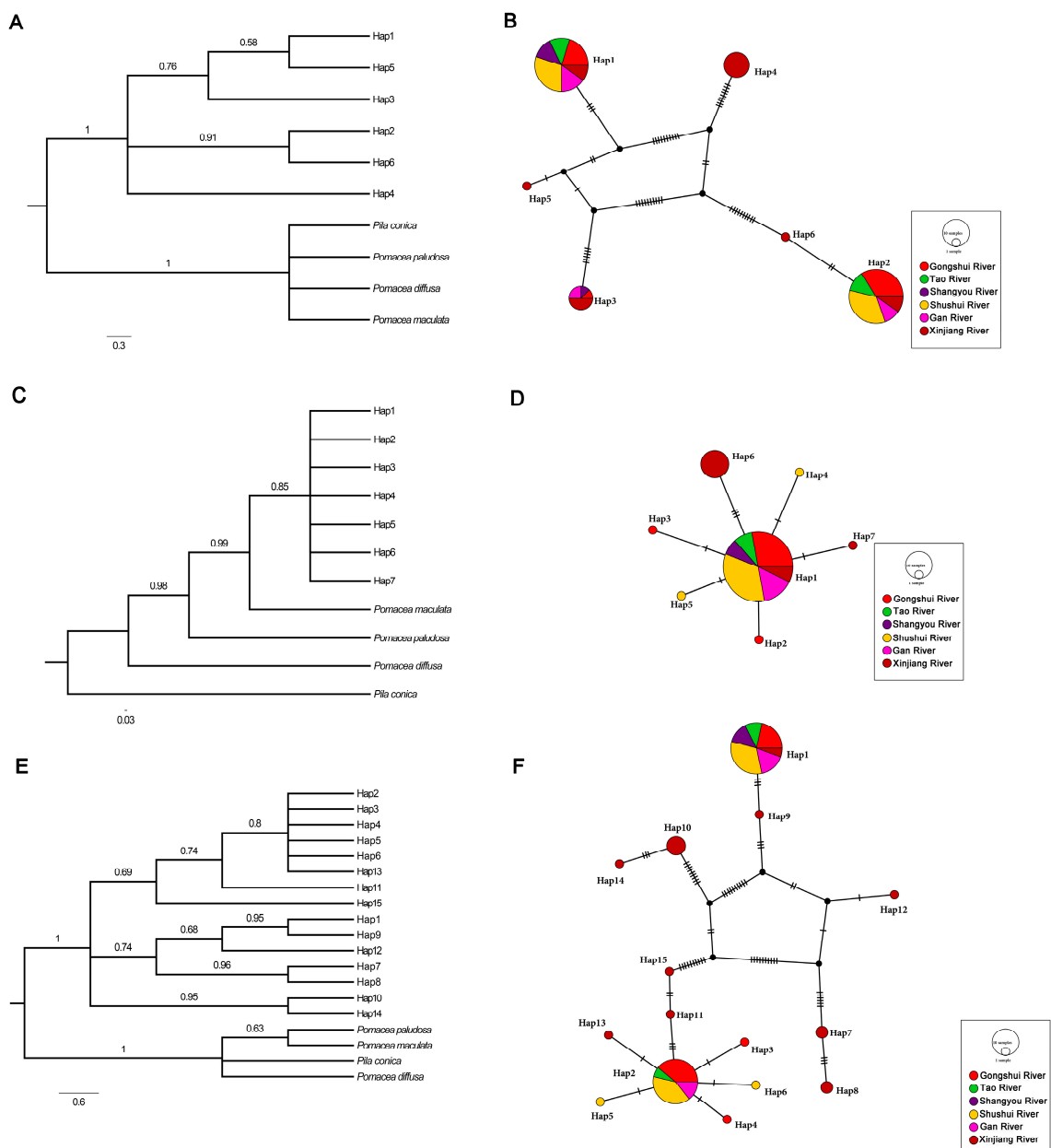

**Figure 2.** Phylogenetic analysis and haplotype network of *P. canaliculata* based on COI (**A**,**B**), 16S (**C**,**D**), and COI+16S (**E**,**F**) datasets.

The results of AMOVA for COI, 16S, and COI+16S datasets showed that 11.46%, 53.79%, and 14.01% of the total genetic variance was among populations, respectively, and the differentiation was both significant ($F_{ST}$ = 0.11, 0.54, and 0.14, respectively; $p < 0.001$; Table 2). Pairwise $F_{ST}$ among the *P. canaliculata* populations of COI, 16S, and COI+16S datasets ranged from 0.0001 to 1.0000, −1.0000 to 1.0000, and 0 to 1.000, respectively, with generally significant values (Table S2).

**Table 2.** Analysis of molecular variation (AMOVA) of *P. canaliculata* based on COI, 16S rRNA, and COI+16S rRNA datasets. Significant results are in bold (* $p < 0.001$).

| Source of Variance | df | Sum of Squares | Variance Components | Percentage of Variation | F-Statistics |
|---|---|---|---|---|---|
| **COI Datasets** | | | | | |
| Among populations | 5 | 112.32 | 0.95 | 11.46 | $F_{ST} = 0.11$ * |
| Within populations | 94 | 688.49 | 7.32 | 88.54 | |
| Total | 99 | 800.82 | 8.27 | 100 | |
| **16S datasets** | | | | | |
| Among populations | 5 | 17.21 | 0.24 | 53.79 | $F_{ST} = 0.54$ * |
| Within populations | 78 | 16.41 | 0.21 | 46.21 | |
| Total | 83 | 33.62 | 0.45 | 100 | |
| **COI+16S datasets** | | | | | |
| Among populations | 5 | 114.13 | 1.18 | 14.01 | $F_{ST} = 0.14$ * |
| Within populations | 78 | 565.47 | 7.25 | 85.99 | |
| Total | 83 | 679.60 | 8.43 | 100 | |

There was a lack of significance of the mismatch distribution, Tajima's D, and Fu's Fs ($p < 0.01$) based on the COI, 16S, and COI+16S datasets, indicating that the apple snail population did not experience demographic expansion (Table 1; Figure 3A,C,E). Additionally, apple snails have had a stable historical population size with a relatively small recent expansion 20,000 to 50,000 years ago based on the BSP of the COI and COI+16S datasets (Figure 3B,D,F).

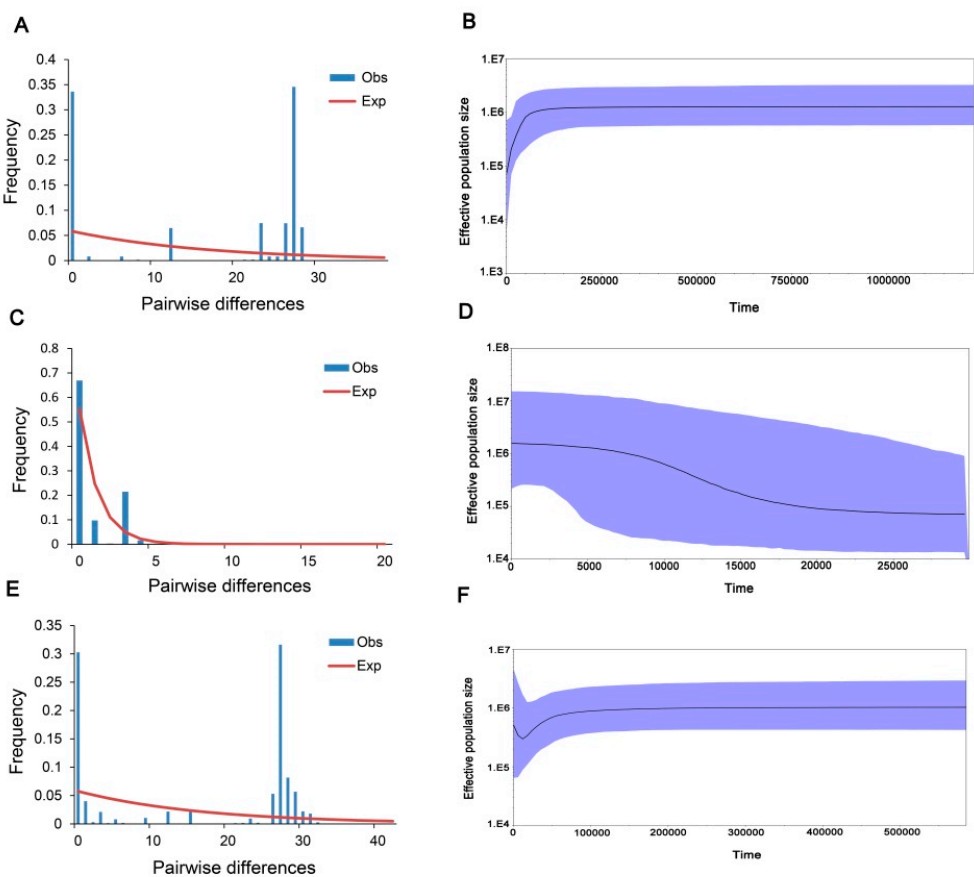

**Figure 3.** Mismatch distribution analysis of six *P. canaliculata* populations and Bayesian skyline plot reconstructing the population size history using an evolutionary rate of $2.0 \times 10^{-8}$ substitutions/site/year based on COI (**A,B**), 16S (**C,D**), and COI+16S (**E,F**) datasets.

### 3.3. Correlation between Genetic and Physicochemical Parameters

Principal coordinate analysis (PCoA) showed that the habitats among the rivers could be broken into five groups based on 11 physiochemical parameters (Figure 4). The first group was the Shangyou River; the second group was the Xinjiang River and the Tao River; the third group was the Shushui River; the fourth group was the Gan River; and the fifth group was the Gongshui River (Figure 4).

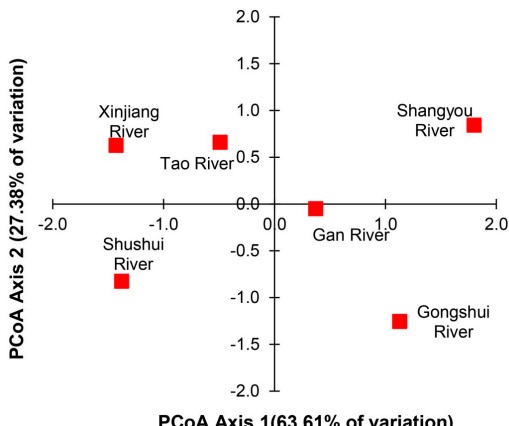

**Figure 4.** Principal coordinates analysis (PCoA) on the dissimilarity of physicochemical parameters across six *P. canaliculata* populations in the Poyang Lake Basin.

The results of redundancy analysis (RDA) showed that 84.2% of the cumulative percentage variance of the genetic diversity–environment variable relations occurred along the first axis, and 100.0% occurred along four axes (Table S3). Haplotype diversity ($H_d$) and nucleotide diversity ($\pi$) of COI and COI+16S datasets were positively correlated with dissolved oxygen concentration, ammonium, and pH. $H_d$ and $\pi$ of 16S datasets were positively correlated with concentration of dissolved oxygen, ammonium concentration, and pH. Genetic diversity indices showed significant correlations with dissolved oxygen and pH based on RDA analysis ($p < 0.05$; Figure 5). Only turbidity and pH were significantly associated with genetic diversity in 16S datasets based on Mantel tests ($p < 0.05$; Table 3).

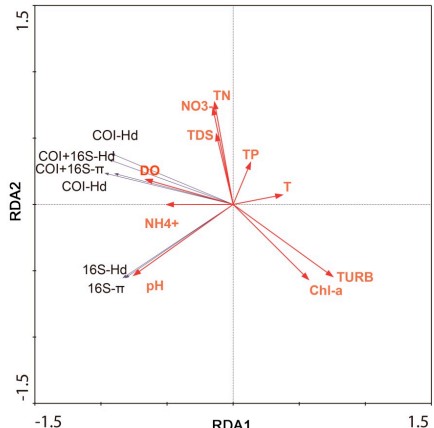

**Figure 5.** Ordination biplot of genetic and physicochemical parameters obtained by redundancy analysis (RDA) across six *P. canaliculata* populations. $H_d$: haplotype diversity; $\pi$: mean nucleotide diversity; DO: dissolved oxygen; TURB: turbidity; T: water temperature; $NH_4^+$: ammonium nitrogen; $NO_3^-$: nitrate nitrogen; TDS: total dissolved solids; Chl-a: chlorophyll-a; TP: total phosphorus; TN: total nitrogen.

**Table 3.** Mantel's correlation coefficients comparing physicochemical parameters and genetic diversity of six *P. canaliculata* populations in the Poyang Lake Basin. Variables include $H_d$: haplotype diversity; $\pi$: nucleotide diversity; DO: dissolved oxygen (mg/L); pH: hydrogen ions; TDS: total dissolved solids (mg/L); TURB: turbidity (NTU); WT: water temperature (°C); Chl-a: chlorophyll-a (mg/L); TN: total nitrogen (mg/L); TP: total phosphorus (mg/L); $NH_4^+$: ammonium nitrogen (mg/L); $NO_3^-$: nitrate nitrogen (mg/L). Significant results are in bold (* $p < 0.05$).

| | COI Datasets | | 16S Datasets | | COI+16S Datasets | |
|---|---|---|---|---|---|---|
| | $H_d$ | $\pi$ | $H_d$ | $\pi$ | $H_d$ | $\pi$ |
| WT | −0.278 | −0.222 | −0.075 | −0.096 | −0.195 | −0.213 |
| TDS | −0.273 | −0.177 | −0.379 | −0.379 | −0.202 | −0.197 |
| pH | 0.384 | 0.089 | **0.881 *** | **0.950 *** | 0.186 | 0.137 |
| DO | 0.250 | 0.379 | −0.221 | −0.231 | 0.351 | 0.369 |
| TURB | 0.396 | 0.702 | **−0.352 *** | **−0.354 *** | 0.645 | 0.692 |
| Chl-a | −0.063 | 0.159 | −0.305 | −0.300 | 0.118 | 0.156 |
| $NH_4^+$ | −0.190 | −0.042 | −0.103 | −0.166 | −0.048 | −0.028 |
| $NO_3^-$ | −0.048 | 0.110 | −0.308 | −0.329 | 0.087 | 0.087 |
| TP | −0.132 | −0.153 | 0.185 | 0.246 | −0.128 | −0.121 |
| TN | 0.016 | 0.239 | −0.287 | −0.275 | 0.237 | 0.245 |

## 4. Discussion

### 4.1. Genetic Diversity of P. canaliculata

Higher levels of genetic diversity among populations could improve the evolutionary potential for dealing with habitat change, effects of pathogen infection, and other selective forces [63,64]. Therefore, knowledge of genetic diversity is important for effectively preventing and managing the spread of invasive species [15,22]. *P. canaliculata* have a significant impact on aquatic ecosystems, such as deflection of indigenous gastropods, water quality changes, and changes in the structure of benthos [7,12]. In China, the distribution of *P. canaliculata* has expanded northward as a consequence of environmental change since its introduction in the 1980s [15,23]. The results showed that while the genetic diversity of *P. canaliculata* in the Poyang Lake Basin was considerable, it is lower than ancestral populations in South America [65]. The differences in genetic diversity may be attributed to possible genetic bottlenecks and genetic drift [35,66], which could be the result of the initial founder event [63]. Moreover, the introduction of alien species to a non-native location may not be directly from the native range but may be from a successful invasive population elsewhere [66,67]. Thus, the genetic diversity of apple snails proposed here may be the result of multiple introduction events and cryptic invasions, which would help control and manage the introduction of apple snail populations.

### 4.2. Genetic Structure of P. canaliculata

Significant genetic differentiation was found among *P. canaliculata* collection sites in the Poyang Lake Basin. The higher levels of genetic differentiation may be attributed to *P. canaliculata* showing a preference for relatively isolated shallow ditches and ponds surrounding human settlements [12,21,34], because geographical isolation affects distribution patterns and genetic structure of species [15,22]. In addition, *P. canaliculata* has relatively limited long distance dispersal abilities, as its movement ability (i.e., crawling) is slow without human assistance [7,12,21,34]. This may lead to a gradual reduction of gene flow, resulting in high genetic differentiation among populations. Moreover, invasive species often experience a bottleneck in their newly colonized habitat if founded by a few individuals, and their genetic variability is expected to decrease in their non-native range [15,21,22,35]. Thus, the genetic structure of apple snails proposed here may be the result of a combination of natural expansion and human-mediated jump dispersal, which would help control and manage the dispersal of apple snail populations.

Apple snails have not had any recent population expansions based on the mismatch distribution and neutrality tests, indicating that population sizes of apple snails have re-

mained quite stable, which may be attributed to apple snails exhibiting high fecundity, rapid growth, and broad physiological tolerances to several abiotic factors [7,12,21]. Additionally, apple snails have had a stable historical population size with a relatively small recent expansion based on the BSP. Hence, constant population size was the best fit for the model to the dataset, with limited support for recent demographic expansion even though it has expanded its global distribution via human-assisted invasion in the last century.

*4.3. Correlation between Genetic and Water Quality Parameters*

The colonization success of invasive species is often related to environmental changes [21]. Abiotic conditions that are beyond tolerances of an organism may caus immediate death or inhibit its ability to establish new populations and become invasive [21]. Some studies showed that the growth and reproduction of *P. canaliculata* are driven by water temperature [30,31], pH [9,20], dissolved oxygen [27], and salinity [28,29]. Our results indicated that genetic variation of *P. canaliculata* was significantly correlated with the concentration of dissolved oxygen and pH ($p < 0.05$). Indeed, a previous study by Glass and Darby (2003) [68] showed that pH predicted snail occurrence and explained variation in survival and growth. Waterbodies with more acidic pH make it extremely difficult for snails to construct shells made of calcium carbonate [69]. Some studies showed that apple snails may be able to tolerate relatively low pH habitats but with reduced growth rates that may negatively impact reproduction [13,70]. For example, the abundance of the closely related species *P. maculata* decreased with lower pH, with snails only found in wetlands with pH > 4.0 [70]. Pierre et al. (2017) [20] showed that snail occurrence was primarily associated with the presence of ditches in wetlands and circumneutral pH (i.e., 6.0–8.0), similar to the pH observed in our study (6.0 to 7.0). Moreover, some studies showed that the distribution and abundance of aquatic snails were related to dissolved oxygen levels [27]. However, due to the amphibious nature of *P. canaliculata* (their possession of both a lung and a gill [7]), dissolved oxygen levels may not greatly influence their distributions [27]. Additionally, their tolerance of low water temperatures is a critical factor in limiting the spread of apple snails, and they did not survive more than 5 days in liquid water at 0 °C [71]. In some cases, there is also a positive correlation between tissue burden and environmental concentration of toxicants for apple snails [7].

Environmental changes (e.g., change of water quality, climate change) may affect genetic patterns of aquatic organisms [72–76]. For example, dam construction could alter hydrological regimes and water quality, resulting in reduced population sizes and isolation among populations, which may indirectly affect the genetic diversity of aquatic organisms [77–80]. Climate change might increase flooding, and rising water levels increase connectivity, which might actually promote dispersal of apple snails and affect genetic diversity [75]. Our study also showed that genetic variation of *P. canaliculata* was affected by differences in water quality. Therefore, these results indicated that differences in water quality affect the snail's survival and persistence.

## 5. Conclusions

Biological invasions are considered one of the biggest threats to biodiversity [4,15]. In this study, *P. canaliculata* among the six collection locations showed robust genetic diversity, significant genetic differentiation, limited gene flow, and stable population dynamics. Genetic diversity in *P. canaliculata* was significantly correlated with concentration of dissolved oxygen and pH.

Additional research using more advanced methods (e.g., genome-wide SNPs) is recommended to further investigate the correlation between genetic and environmental variation and how this might influence colonization success and to prevent the invasion and spread of non-native species in other habitats. Research on the effects of environmental variables on genetic diversity and structure and their effects on invader persistence and colonization remains crucial. Our results will provide a basis for better understanding how to prevent and manage the spread of invasive *P. canaliculata*.

**Supplementary Materials:** The following supporting information can be downloaded at: https://www.mdpi.com/article/10.3390/d15101048/s1, Table S1 Mean water physicochemical parameters from six *P. canalicuata* populations in the Poyang Lake Basin collected October 2016 and January, May and August 2017. DO: dissolved oxygen concentration (mg/L), pH: concentration of hydrogen ions, TURB: turbidity (NTU), T: water temperature ($^\circ$C), $NH_4^+$: ammonium concentration (mg/L), $NO_3^-$: nitrate concentration (mg/L), TDS: total dissolved solids (mg/L), Chl-a: chlorophyll-a concentration (units?), TP: total phosphorus (mg/L), TN: total nitrogen (mg/L). The value of all parameters were mean $\pm$ standard error; Table S2 Analysis of genetic differentiation coefficient (Fst) calculated using COI, 16S and COI+16S datasets among six populations of *P. canaliculata*. Bold type indicates statistical significance ($p < 0.001$). GS: Gongshui River; TR:Tao River; SY: Shangyou River; SS: Shushui River; GR: Gan River; XR: Xinjiang River; Table S3 Summary statistics for the first 4 axes of RDA performed between physicochemical parameters and genetic parameters.

**Author Contributions:** X.L., Y.Z., S.O. and X.W. conceived the study. All authors contributed to the study design and data collection. X.L. and Y.Z. analyzed the data. X.L., Y.Z., S.O. and X.W. led the writing of the manuscript. All authors have read and agreed to the published version of the manuscript.

**Funding:** This research was funded by the National Key R & D Program of China (2018YFD0900801), Science and Technology Planning Project of Meizhou, China (2021B0201001), National Key Research and Development Program of China (No. 2016YFC1202000, 2016YFC1202002), Educational Commission of Guangdong Province of China (2021ZDZX4054), Guangdong Provincial Science & Technology Innovation and Rural Revitalization Strategy (2021A0305) and Innovation and Entrepreneurship Project of Jiaying University (622A03065). And The APC was funded by Science and Technology Planning Project of Meizhou, China (2021B0201001) and National Key R & D Program of China (2018YFD0900801).

**Institutional Review Board Statement:** Not applicable.

**Data Availability Statement:** All sequences were deposited in the NCBI (National Center of Biotechnology Information; https://www.ncbi.nlm.nih.gov/ (accessed on 1 September 2020)) under GenBank accession numbers MH602328–MH602347.

**Acknowledgments:** This work is supported by grants from the National Key R & D Program of China (2018YFD0900801), Science and Technology Planning Project of Meizhou, China (2021B0201001), National Key Research and Development Program of China (No. 2016YFC1202000, 2016YFC1202002), Educational Commission of Guangdong Province of China (2021ZDZX4054), Guangdong Provincial Science & Technology Innovation and Rural Revitalization Strategy (2021A0305) and Innovation and Entrepreneurship Project of Jiaying University (622A03065). The authors report no conflicts of interest. The authors alone are responsible for the content and writing of this article.

**Conflicts of Interest:** The authors report that they have no conflict of interest.

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
