# Peer review of "Environmental Correlates to Genetic Diversity and Structure in Invasive Apple Snail (Pomacea canaliculata) Populations in China"

_diversity, doi:10.3390/d15101048_

Round 1

Reviewer 1 Report

Overall the paper is certainly publishable.

(1) Some attention needs to be given to minor plagiarism

(2) Authors have a tendancy to make statements and provide no explanation of the concepts or the examples behind them. Evidence and explanation is required for all statements. For example you cannot simply say climate change might affect apple snail invasion success - you need to explain why that might be and give examples. This happens frequently in the intro and discussion

(3) results are a bit confusing, mostly because statements of significant correlations are made with no evidence - show your work, provide the stats, even if its in a supplemental table, so the readers understand how you came to your conclusions.

I would also love a bit more detail on the goals - why does understanding how water quality parameters affect or link with genetic variation help management? please provide some more detail and maybe an example where this is true elsewhere to help the reader with context.

Revisions are not really major (but more than minor revisions), some rephrasing and context, providing detail and examples and being more explicit. But the bones of the paper are fine.

Quality of English was fine, some small corrections for clarity or to pluralize certain terms, but very comprehensible and easy to read. Well done.

Author Response

POINT-BY-POINT RESPONSE

Dear Ms. Libby Liu,

Please find attached the revised version of our manuscript, “Environmental Corrlates to Genetic Diversity and Structure in Invasive Apple Snail (Pomacea canaliculata) Populations in China” (MS number: diversity-2581618), submited to Diversity. We have carefully considered all comments, and we now submit the new version. In general, we address each specific comment but for those particular cases where not, we explain our reason for doing so.

Best regards,

Xiongjun Liu

on behalf of the authors. 

Point-by-point rebuttal                                                              

Editor and Reviewer comments

Reviewer #1

Comments and Suggestions for Authors

Overall the paper is certainly publishable.

(1) Some attention needs to be given to minor plagiarism

Authors: This was revised.

(2) Authors have a tendancy to make statements and provide no explanation of the concepts or the examples behind them. Evidence and explanation is required for all statements. For example you cannot simply say climate change might affect apple snail invasion success - you need to explain why that might be and give examples. This happens frequently in the intro and discussion

Authors: This was revised.

(3) results are a bit confusing, mostly because statements of significant correlations are made with no evidence - show your work, provide the stats, even if its in a supplemental table, so the readers understand how you came to your conclusions.

Authors: This was revised.

I would also love a bit more detail on the goals - why does understanding how water quality parameters affect or link with genetic variation help management? please provide some more detail and maybe an example where this is true elsewhere to help the reader with context.

Authors: This was revised. Higher levels of genetic diversity among populations could improve the evolutionary potential for dealing with habitat change, effects of pathogen infection, and other selective forces. P. canaliculata has relatively limited long distance dispersal abilities as movement ability (i.e., crawling) is slow without human assistance. This may lead to a gradual reduction of gene flow, and resulting in high genetic differentiation among populations. Moreover, invasive species often experience a bottleneck in their newly colonized habitat if founded by a few individuals, and their genetic variability is expected to decrease in their non-native range. Several different haplotypes of P. canaliculata present at the same site, as well as different combinations of haplotypes between different sites, indicating that these haplotypes are secondary and multiple introductions. Therefore, knowledge of the genetic diversity is important for effectively preventing and managing the spread of invasive species.

Revisions are not really major (but more than minor revisions), some rephrasing and context, providing detail and examples and being more explicit. But the bones of the paper are fine.

Authors: This was revised.

Reviewer 2 Report

The authors present a study on the relationship between environmental variables and the genetic diversity of an invasive gastropod species in China. I consider the work to be original with interesting results, but I believe the authors should make some adjustments before the manuscript can be considered for publication.

An important issue concerns the sampled locations and the physicochemical parameters measured at different times of the year. It is not clear in the text (and I did not have access to the supplementary material) whether these variables were measured throughout the months mentioned by the authors or if each point was measured four times (once per mentioned month). Depending on this, the results could be affected.

Regarding the molecular analyses, I ask the authors why they did not consider other sequences of P. canaliculata from South America or from any invaded area. Why were two species from the same genus considered in the analyses instead of another genus within the Ampulariidae family?

 In the Results section (Correlation between genetic and physicochemical parameters), all the information presented does not come directly from the PCoA analysis but from tables with records of maximum and minimum values of physicochemical parameters for each river. Furthermore, the graph presented in Figure 4 does not show the variables nor does it correspond to the information in the paragraph to which it is referenced.

In the Discussion, I missed an effective comparison of the authors' findings with previous studies. On the other hand, since the authors attempt to relate environmental variables, the discussion does not mention important issues such as connectivity between the sampled rivers, the presence of anthropogenic impacts in the basin, and how the implementation of a dam in Lake Poyang would affect the distribution of P. canalicutata.

Other comments are directly included in the attached file.

I'm not a native English speaker, but at various points during the reading, I experienced some difficulty with the clarity of the manuscript's writing (phrases and terms used). I suggest that the authors conduct a thorough grammar review in the new version.

Author Response

POINT-BY-POINT RESPONSE

Dear Ms. Libby Liu,

Please find attached the revised version of our manuscript, “Environmental Corrlates to Genetic Diversity and Structure in Invasive Apple Snail (Pomacea canaliculata) Populations in China” (MS number: diversity-2581618), submited to Diversity. We have carefully considered all comments, and we now submit the new version. In general, we address each specific comment but for those particular cases where not, we explain our reason for doing so.

Best regards,

Xiongjun Liu

on behalf of the authors. 

Point-by-point rebuttal                                                              

Editor and Reviewer comments

Reviewer #2

The authors present a study on the relationship between environmental variables and the genetic diversity of an invasive gastropod species in China. I consider the work to be original with interesting results, but I believe the authors should make some adjustments before the manuscript can be considered for publication.

An important issue concerns the sampled locations and the physicochemical parameters measured at different times of the year. It is not clear in the text (and I did not have access to the supplementary material) whether these variables were measured throughout the months mentioned by the authors or if each point was measured four times (once per mentioned month). Depending on this, the results could be affected.

Authors: Eleven physicochemical parameter variables at six rivers during October 2016 and January, May and August 2017 were measured four times in the study area. The data of water quality data comes from Xing et al. (2019) and Li et al. (2019).

Regarding the molecular analyses, I ask the authors why they did not consider other sequences of P. canaliculata from South America or from any invaded area. Why were two species from the same genus considered in the analyses instead of another genus within the Ampulariidae family?

Authors: The goal of this study is not to focus on taxonomy. We analyzed the correlation between a suite of physicochemical parameter variables and the genetic diversity of P. canaliculata. The data of physicochemical parameter variables from South America or from any invaded area are lacking, so we can′t analyze the correlation between physicochemical parameter variables and the genetic diversity.

In the Results section (Correlation between genetic and physicochemical parameters), all the information presented does not come directly from the PCoA analysis but from tables with records of maximum and minimum values of physicochemical parameters for each river. Furthermore, the graph presented in Figure 4 does not show the variables nor does it correspond to the information in the paragraph to which it is referenced.

Authors: This was revised.

In the Discussion, I missed an effective comparison of the authors' findings with previous studies. On the other hand, since the authors attempt to relate environmental variables, the discussion does not mention important issues such as connectivity between the sampled rivers, the presence of anthropogenic impacts in the basin, and how the implementation of a dam in Lake Poyang would affect the distribution of P. canalicutata.

Authors: This was revised.

Other comments are directly included in the attached file.

Authors: This was revised.

Round 2

Reviewer 1 Report

The authors have made some attempt to address my original comments. But some problems still remain. Minor revisions (but essential). 

Please be careful about lifting sentences from other papers - you MUST paraphrase. 

Some more detail on methods

A better explanation on how their results will help control and manage populations is necessary. This is apparently the goal of the paper, but not explanation on how that would work is provided. how does knowing which environemental variables drive genetic variation inform control? how could we use this information for management???? 

Language is understandable and fairly well written. A few small corrections will be needed by the copy editor.

Author Response

POINT-BY-POINT RESPONSE

Dear Ms. Libby Liu,

Please find attached the revised version of our manuscript, “Environmental Corrlates to Genetic Diversity and Structure in Invasive Apple Snail (Pomacea canaliculata) Populations in China” (MS number: diversity-2581618), submited to Diversity. We have carefully considered all comments, and we now submit the new version. In general, we address each specific comment but for those particular cases where not, we explain our reason for doing so.

Best regards,

Xiongjun Liu

on behalf of the authors. 

Point-by-point rebuttal                                                              

Editor and Reviewer comments

Reviewer #1

The authors have made some attempt to address my original comments. But some problems still remain. Minor revisions (but essential).

Please be careful about lifting sentences from other papers - you MUST paraphrase.

Authors: This was revised.

Some more detail on methods

Authors: This was revised.

A better explanation on how their results will help control and manage populations is necessary. This is apparently the goal of the paper, but not explanation on how that would work is provided. how does knowing which environemental variables drive genetic variation inform control? how could we use this information for management????

Authors: This was explained in Conclusion.

Reviewer 2 Report

In the revised version of the Manuscript, the authors have carefully considered the observations and suggestions made in both the text and the figures. I have no further changes or modifications to request.

Author Response

POINT-BY-POINT RESPONSE

Dear Ms. Libby Liu,

Please find attached the revised version of our manuscript, “Environmental Corrlates to Genetic Diversity and Structure in Invasive Apple Snail (Pomacea canaliculata) Populations in China” (MS number: diversity-2581618), submited to Diversity. We have carefully considered all comments, and we now submit the new version. In general, we address each specific comment but for those particular cases where not, we explain our reason for doing so.

Best regards,

Xiongjun Liu

on behalf of the authors. 

Point-by-point rebuttal                                                              

Editor and Reviewer comments

Reviewer #2

In the revised version of the Manuscript, the authors have carefully considered the observations and suggestions made in both the text and the figures. I have no further changes or modifications to request.

Authors: Thanks for your suggestion.
